# GUIDEEDIT: ENHANCING FACE VIDEO EDITING WITH FINE-GRAINED CONTROL

## ABSTRACT

Face video editing (FVE) requires maintaining temporal consistency and identity preservation while manipulating specific attributes. However, existing FVE methods often introduce unwanted artifacts and affect non-target attributes during editing. To address these limitations, we propose **GuideEdit** to enhance the precision of face video editing. Given the inherent linearity of the latent variables in the bottleneck layer of the diffusion U-Net model, there exists a linear mapping between the input and the latent representation. This allows us to extract a latent basis within the latent space that effectively encodes the key features related to target facial attributes. By comparing the latent basis of the original video to that of the manipulated video, we quantify the manipulation degree, which indicates the extent of changes made. This manipulation degree serves as a guide for determining the specific components to be edited, then we achieve more precise control at each denoising step. Integrating this fine-grained control into the editing process allows GuideEdit to enhance temporal consistency and preserve identity of FVE, while minimizing the introduction of artifacts. Extensive experiments on diverse real-world videos demonstrate the effectiveness of GuideEdit, showcasing its ability to achieve precise, high-quality edits that maintain coherence across frames and ensure the preservation of essential visual elements.

## 1 INTRODUCTION

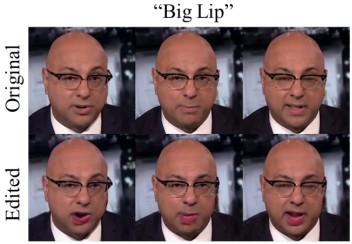 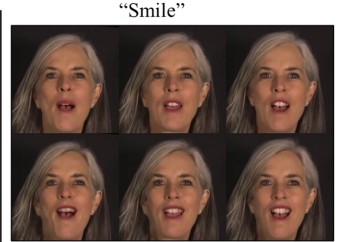 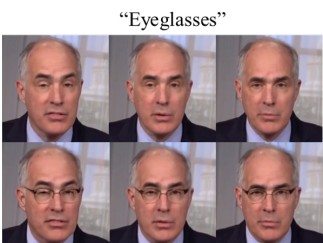

Figure 1: Given the editing direction, the proposed GuideEdit is able to edit real-world face videos without affecting the identity and the background, while ensuring smooth transitions over time.

Face attribute editing has emerged as an essential task in computer vision, with applications ranging from film production to virtual reality, social media content, and digital avatars (Zhan et al., 2023; Kim et al., 2023; Yao et al., 2021; Zhang et al., 2018a; Zhu et al., 2020). While significant progress has been made in face image editing (Shen et al., 2020; Zhu et al., 2020; Wang et al., 2022), comparatively fewer efforts have focused on FVE. The core challenge in FVE is to modify specific facial attributes (*i.e.*, expression, age or hairstyle) while maintaining the temporal consistency, identity preservation, and background integrity of the video (Wang et al., 2024). Traditional image-based editing methods can't be applied to video editing directly, because they struggle to maintain consistency across video frames due to the complex temporal dependencies and the intricate relationship between facial attributes and identity (Ceylan et al., 2023).

Several GAN-based methods for FVE utilize pre-trained StyleGAN models (Tzaban et al., 2022; Patashnik et al., 2021; Karras et al., 2019; Shen et al., 2020) to facilitate the editing process. These approaches commonly employ GAN inversion(Karras et al., 2020; Xia et al., 2022), where the pre-trained GAN is used to map the input video frames into a latent space, enabling the application of

desired edits. However, the quality of the edited video is heavily reliant on the effectiveness of the GAN inversion. These GAN-based methods often struggle to accurately reconstruct the original input, resulting in suboptimal editing quality(Preechakul et al., 2022). More recently, diffusion models renowned for their strong generative capabilities, have demonstrated success in FVE (Kim et al., 2023; Preechakul et al., 2022), outperforming GAN-based approaches in editing quality. The editing process in diffusion-based FVE methods is typically framed as a conditional generation task (Zhang et al., 2023; Croitoru et al., 2023), where the desired target attribute is progressively introduced into the video at various stages of the denoising process (Kim et al., 2023). However, simply introducing the target attribute at different denoising steps without additional constraints can inadvertently affect the other attributes of the video, such as identity, expression, or background. This occurs because the diffusion model lacks precise control over the editing process (Zhao et al., 2024; Yu et al., 2023), leading to undesired modifications in non-target regions or features.

To improve diffusion-based FVE and achieve precise control, we propose GuideEdit that edits real-world face videos without affecting the identity and background features, while ensuring time consistency (as presented in Figure 1). Given the local linearity of the latent variables in the bottleneck layer of the UNet architecture (Park et al., 2023; Kwon et al., 2022), a linear mapping exists between the inputs and the latent variables. However, since the latent variables encode both the input frame features and the assigned attributes features, directly using them to measure the impact of target attributes could result in interference from unrelated components (Park et al., 2023). Therefore, GuideEdit leverages the local linearity property to isolate and extract only the latent basis vectors that are most relevant to the target attributes, avoiding unintended modifications to other elements. To ensure precise control, GuideEdit corrects the directional deviation of the estimated noise between the input with the introduced attribute and the original input according to the similarity between the latent basis of the newly introduced target attributes and the original video. This correction refines the denoising process to focus exclusively on the components associated with target attributes, ensuring that only the target attribute is modified while preserving other attributes. As a result, the effectiveness of the manipulation process is significantly enhanced, allowing for more precise and consistent editing without compromising the integrity of the original video.

We summary the contributions of our proposed method shortly as follows.

- We propose a new approach GuideEdit for FVE within the diffusion model framework, where precise control is achieved by leveraging the local linearity of the latent variables in the bottleneck layer of a UNet architecture.

- We introduce a latent basis extraction mechanism that identifies the most influential features of the input video's conditions. By calculating the similarity between the latent basis of the original and edited video, we quantify the degree of modification, providing a precise control signal for the editing process.

- We present a proximal guidance mechanism that uses the latent basis similarity to guide the denoising process in the diffusion model. This ensures that changes are confined to the specified target attribute, reducing unintended alterations and enhancing the quality of the edited video.

- Extensive experiments on real-world datasets demonstrate the effectiveness of the proposed method, showing improvements in identity preservation, target attribute modification, and temporal consistency.

## 2 RELATED WORK

### 2.1 FACE VIDEO EDITING

Existing methods for FVE can be broadly categorized into two types: GAN-based and diffusion-based methods. GAN-based methods typically leverage pre-trained GAN models like Style-GAN (Tzaban et al., 2022; Karras et al., 2019) for face video manipulation. A common technique in these methods involves GAN inversion, where the input video frames are mapped to the latent space of a pre-trained GAN (Karras et al., 2020; Xia et al., 2022), and the desired edits are applied by manipulating the latent codes (Patashnik et al., 2021; Shen et al., 2020). While GAN-based methods have achieved high-quality image synthesis, they suffer from several drawbacks in the context of

video editing. The effectiveness of GAN-based video editing is heavily dependent on the quality of the GAN inversion, which often struggles to perfectly reconstruct the input video, leading to loss of detail or failure to preserve identity features (Preechakul et al., 2022).

In diffusion-based methods, the editing process is typically formulated as a conditional generation task, where the target attribute is introduced into the video during the denoising process (Kim et al., 2023). These models gradually modify the video by reversing a noising process, progressively refining the video's attributes over several steps. Diffusion-based methods offer several advantages over GAN-based approaches. Due to their iterative nature, diffusion models can more effectively preserve temporal consistency, as the modifications are made gradually, and the generative process considers the entire video context. However, these methods typically involve a trade-off in terms of computational cost, as the iterative denoising steps are time-consuming, leading to slower inference times compared to GAN-based methods.

## 2.2 LATENT SPACE ANALYSIS

The study of latent spaces has gained significant attention in recent years. In the field of Generative Adversarial Networks (GANs), researchers have proposed various methods to manipulate the latent space to achieve the desired effect in the generated images (Ramesh et al., 2018; Patashnik et al., 2021; Abdal et al., 2021; Shen & Zhou, 2021; Härkönen et al., 2020). More recently, several studies have examined the geometrical properties of latent space in GANs and utilized these findings for image manipulations (Choi et al., 2021; Zhu et al., 2021). Some studies have applied Riemannian geometry to analyze the latent spaces of deep generative models (Arvanitidis et al., 2017; 2020; Chen et al., 2018; Lee & Park, 2023; Lee et al., 2022; Shao et al., 2018). (Shao et al., 2018) proposed a pullback metric on the latent space from image space Euclidean metric to analyze the latent space's geometry. This method has been widely used in VAEs and GANs because it only requires a differentiable map from latent space to image space. And (Park et al., 2023) extend it into diffuison models (DMs) to investigate the geometry of latent space of DMs to facilitate the image editing. However, it is challenging for the pullback metric to accurately capture the geometry of the latent space from the image space, as the image space contains excessive information, making it difficult to identify the correct directions for editing.

## 2.3 INVERSION-BASED GUIDANCE

DDIM inversion (Song et al., 2020) exhibits great potential in editing tasks by deterministically calculating and encoding the context information in a latent and reconstructing the original image with it. Applying editing prompt upon the inverted latent code to guide the denoising process greatly improved the test-time efficiency. Leveraging optimization on null-text embedding, Null-text Inversion (Mokady et al., 2023) further improved the identity preservation of the edit. However, all these methods rely on optimization at test-time for accurate reconstruction, which typically requires several minutes. Negative-prompt inversion (NPI) (Miyake et al., 2023) further reduces the computation cost for the inversion step while generates similarly competitive reconstruction results as Null-text inversion. However, NPI may occasionally introduce artifacts due to its underlying assumptions. And ProxEdit (Han et al., 2024) introduces an inversion guidance technique that applies a one-step gradient descent on the current latent representation, aligning it with the inversion latent to correct errors introduced during the reconstruction process. However, this ProxEdit method requires manually setting correction thresholds for different editing tasks, which can introduce additional bias.

## 3 DIFFUSION-BASED FACE VIDEO EDITING

Let $X = \{x_1, ..., x_n\}$ represent a video consisting of $n$ frames, where each $x_i$ is a single frame from the original video. The goal of diffusion-based human video editing is to manipulate specific attributes of the human subjects in the video (*i.e.*, facial expressions, hairstyles) while preserving other attributes such as identity, background, and temporal consistency. The editing process in diffusion-based methods can be formulated as a conditional generation task, where target attributes are encoded as conditioning inputs and introduced during the video reconstruction process.

The video frames are firstly reversed into noisy representations by forward diffusion process. Then the forward diffusion process progressively applies noise to the input frames, resulting in noisy rep-

resentations $X_t = \{x_{t,1}, ..., x_{t,n}\}$ for each time step $t \in T$, where $T$ is the total number of diffusion steps. The reverse diffusion process reconstructs the video by gradually removing the noise, and during this process, the target attribute encoded as a condition $\Delta c$ is introduced at denoising steps. The reverse process is typically parameterized by a neural network $\mathcal{F}_\theta$ with parameters $\theta$ that predicts the noise in each frame, guiding the denoising process:

$$p_\theta(X_{t-1}|X_t, \Delta c) = \mathcal{N}(X_{t-1}; \mu_\theta(X_t, t, \Delta c), \Sigma_\theta(X_t, t, \Delta c)) \tag{1}$$

Thus, the final output video $\hat{X} = X_0 = \{\hat{x}_1, ..., \hat{x}_n\}$ retains the introduced attribute, while preserving identity and background details.

However, while diffusion-based FVE introduces target attributes effectively, it struggles to preserve identity and background details due to the lack of precise control in the editing process. To address this limitation, we introduce GuideEdit in Section 4, which enhances the accuracy and quality of diffusion-based FVE.

## 4 METHOD: GUIDEEDIT

We propose an effective diffusion-based human video editing method, GuideEdit, with its framework illustrated in Figure 2. The key components of GuideEdit are outlined in the following sections: the forward diffusion process is detailed in Section 4.1, the latent basis extraction is described in Section 4.2, and the proximal guidance mechanism is introduced in Section 4.3.

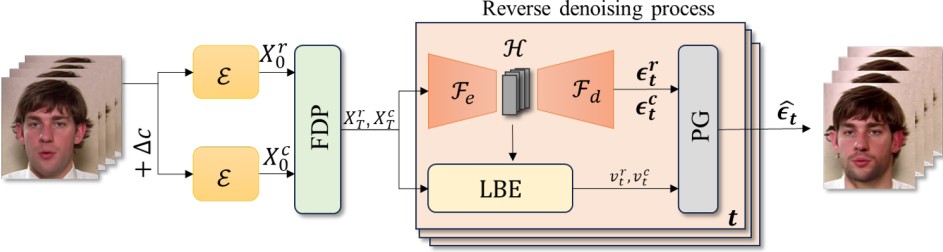

Figure 2: **The framework of proposed GuideEdit.** (a). The proposed GuideEdit utilizes a forward diffusion process (FDP) module (refer to Section 4.1) to reverse the encodings of both the original video and the manipulated video with the specified attribute. (b). The reversed encodings are then fed into a UNet-based noise estimator. The latent basis is subsequently extracted using the latent basis extraction (LBE) module (refer to Section 4.2). (c). The similarity between the latent basis is computed, and the proximal guidance (PG) module (refer to Section 4.3) leverages this similarity to guide the editing direction, ensuring high-quality manipulation of the video.

### 4.1 FORWARD DIFFUSION PROCESS

We present the process of encoding the input $X$ into $X_0^c$ in Figure 3. To encode the conditions related to the target attribute into the video, we first obtain the embedding for the original frames using a pre-trained condition generator, denoted as $\mathcal{E}_c$: $c^r = \mathcal{E}_c(X)$. Next, we utilize a pre-trained encoder $\mathcal{E}_e$ to jointly encode the video frames and the associated embedding into conditions (the process of obtaining $\Delta c$ can refer to Appendix C.2), which are then used as conditions during the denoising process:

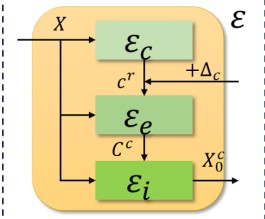

$$\mathcal{C}^r = \mathcal{E}_e(X, c^r), \ \mathcal{C}^c = \mathcal{E}_e(X, c^r + \Delta c) \tag{2}$$

where $\mathcal{C}^r$ and $\mathcal{C}^c$ are utilized as conditions for the denoising of the original and manipulated frames, respectively. And the input representations at time step $t = 0$ are derived using a frozen input encoder $\mathcal{E}_i$: $X_0^r = \mathcal{E}_i(X, \mathcal{C}^r)$ and $X_0^c = \mathcal{E}_i(X, \mathcal{C}^c)$, $X_0^r$ represents the original input representation and $X_0^c$ serves as the conditional input representation for manipulation.

Figure 3: The architecture of encoder $\mathcal{E}$, consists of $\mathcal{E}_c$, $\mathcal{E}_e$ and $\mathcal{E}_i$.

After obtaining the encoded input representations $X_0^r$, $X_0^c$, the forward diffusion can be applied:

$$q(X_t^r|X_0^r) = \mathcal{N}(X_t^r; \sqrt{\alpha_t}X_0^r, (1-\alpha_t)\epsilon_t^r), \ \epsilon_t^r = \mathcal{F}_\theta(X_0^r, t, \mathcal{C}^r) \tag{3}$$

where $\mathcal{F}_\theta$ denotes a pre-trained noise estimator, and $X_t^r$ represents the noisy representation at diffusion step $t$. The parameter $\alpha_t$ controls the noise scale at step $t$. Through this process, $X_T^r$ is generated by the forward diffusion process. Similarly, the forward diffusion process is applied to $X_0^c$ to obtain $X_T^c$.

## 4.2 LATENT BASIS EXTRACTION

The noisy representations $X_T^r$ and $X_T^c$ obtained in Section 4.1 are put into a pre-trained UNet $\mathcal{F}$ to predict the noise of each frame, we use $\mathcal{F}_e$ and $\mathcal{F}_d$ to denote the encoder and decoder of the UNet respectively. Since the extraction of the latent basis is identical for both $X_T^r$ and $X_T^c$, we use $X_T^c$ as an example for simplicity. To streamline the presentation, we let $\mathcal{X}$ represent $X_t^c$, $\mathcal{H}$ denote the latent variable, and $\mathcal{C}$ represent $\mathcal{C}^c$ at time step $t$.

The latent variable $\mathcal{H}$ in the bottleneck layer of the U-Net has been shown to exhibit a locally linear structure (Kwon et al., 2022), which makes it suitable for using the Euclidean metric to measure changes in $\mathcal{H}$ (Kim et al., 2023). In the denoising process, the transformation from the input representations to the latent space can be expressed as $\mathcal{F}_e : \mathcal{X}, \mathcal{C} \to \mathcal{H}$, where $\mathcal{F}_e$ maps the input $\mathcal{X}$ and the editing conditions $\mathcal{C}$ to the latent variable $\mathcal{H}$.

However, since $\mathcal{X}$ contains a lot of information unrelated to the specific editing direction, the variability it introduces into $\mathcal{H}$ might not align with the desired editing directions. To overcome this issue, we focus primarily on how $\mathcal{C}$ (the editing condition) influences $\mathcal{H}$, effectively isolating the impact of the target attribute from other unrelated aspects of $\mathcal{X}$. This approach enables us to better control the editing process by only adjusting the components of $\mathcal{H}$ that are relevant to the intended changes, ensuring more precise and consistent video edits.

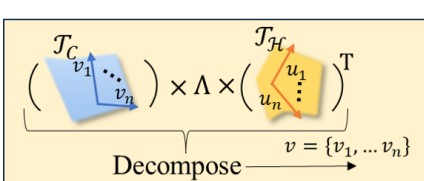

Figure 4: The illustration of extracting the latent basis.

Since the video editing process incorporates the additional condition $\mathcal{C}$ into the denoising steps, $\mathcal{C}$ directly influences key features in the latent space $\mathcal{T}_\mathcal{H}$, where $\mathcal{T}_{(.)}$ denotes the vector space. Therefore, our goal is to identify the local latent vectors $V = \{v_1, \ldots, v_n\} \in \mathcal{T}_\mathcal{C}$ that exhibit significant variability within the tangent space of the latent variable $\mathcal{H}$, denoted as $\mathcal{T}_\mathcal{H}$. By focusing on these local latent vectors, we can effectively capture the key aspects of the editing direction that drive changes in the latent space, ensuring that the manipulation of the video aligns with the intended attribute modifications while preserving other important details such as identity and background.

The linear relationship between $\mathcal{C}$ and $\mathcal{H}$ can be expressed as a linear map: $\mathcal{T}_\mathcal{C} \to \mathcal{T}_\mathcal{H}$. This linear transformation is described by the Jacobian matrix $J_\mathcal{C}$, which captures how a vector $v \in \mathcal{T}_\mathcal{C}$ is mapped to a vector $u \in \mathcal{T}_\mathcal{H}$ through the relation $u = J_\mathcal{C} v$. Given the local linearity of $\mathcal{H}$ in the latent space, the pullback of $\mathcal{H}$ allows us to assign a meaningful geometric structure to $\mathcal{C}$, enabling more precise control over the editing process by understanding how changes in $\mathcal{C}$ affect the latent space $\mathcal{H}$, the norm of $v$ can be measured:

$$||v||_{pb}^2 = <u, u>_\mathcal{H} = v^\top J_\mathcal{C}^\top J_\mathcal{C} v \tag{4}$$

where $< u, u >_\mathcal{H} = u^\top u$ is the dot product of $u$ defined in the Euclidean space with the local linearity of $\mathcal{H}$.

The vectors $V = \{v_1, \ldots, v_n\} \in \mathcal{T}_\mathcal{C}$ that maximize $||v||_{pb}^2$ can be derived through the singular value decomposition (SVD) of the Jacobian matrix $J_\mathcal{C} = U \Lambda V^\top$, as illustrated in Figure 4. Here, $V = \{v_1, \ldots, v_n\}$ represents the right singular vectors of $J_\mathcal{C}$, $U = \{u_1, \ldots, u_n\} \in \mathcal{T}_\mathcal{H}$ represents the left singular vectors, and $\Lambda$ is a diagonal matrix of singular values, it has $J_\mathcal{C} v_i = \Lambda_i u_i$. The extracted latent basis vectors $V = \{v_1, \ldots, v_n\}$ correspond to directions in the latent space that are highly responsive to the conditions encoded in $\mathcal{C}$, offering key insights into how the video editing process responds to specific attributes. Henceforth, we obtain the latent basis responses corresponding to the conditions $\mathcal{C}^r$ and $\mathcal{C}^c$, denoted as $V^r = \{v_1^r, \ldots, v_n^r\}$ and $V^c = \{v_1^c, \ldots, v_n^c\}$, respectively. The similarity between these latent basis vectors $V^r$ and $V^c$ can be measured through using a cosine

similarity metric, defined as:

$$S_{\mathcal{C}}(V^r, V^c) = \cos^{-1}(\phi)/\pi, \ \cos(\phi) = \frac{1}{n} \sum_{i=1}^{n} \frac{v_i^r v_i^c}{||v_i^r|| ||v_i^c||} \tag{5}$$

This similarity quantifies the degree of alignment between the original and manipulated conditions, providing a means to assess the extent of changes introduced during the editing process.

### 4.3 PROXIMAL GUIDANCE

The latent basis associated with different conditions is extracted as described in Section 4.2, and the similarity between $V^r$ and $V^c$ can be utilized to provide more precise guidance for video manipulation. We denote the computed similarity as $a = S_{\mathcal{C}}(V^r, V^c)$, refer to Equation 5. This similarity $a$ serves as a key factor in adjusting the manipulation process, ensuring that only the target attributes are modified while preserving other important characteristics like identity and background.

FVE is achieved by introducing conditions into the denoising process, but these introduced conditions can lead to inaccurate reconstructions. As shown in Figure 5, due to the absence of precise guidance, the direction of $\epsilon^c$ deviates significantly from the original direction $\epsilon^r$, which results in errors during the editing process, such as an inability to preserve the identity and background information of the video. Given that the similarity $a = S_{\mathcal{C}}(V^r, V^c)$ measures the impact of the conditions on the model, we propose using this similarity as guidance to regulate the denoising process.

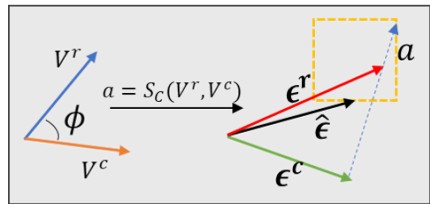

Figure 5: The illustration of proximal guidance.

To ensure that the directions of $\epsilon^c$ and $\epsilon^r$ remain consistent with the similarity $a$, it is crucial that only the target attribute is manipulated during the editing process. To achieve this, we employ a dynamic threshold rather than a fixed one. Specifically, we select the $1 - a$ quantiles from the matrix $|\epsilon^c - \epsilon^r|$ and denote the cutoff value as $\lambda$. This allows us to obtain the following matrix:

$$\mathcal{M} = |\epsilon^c - \epsilon^r| < \lambda, \ \hat{\epsilon} = \epsilon^c + M \odot (\epsilon^r - \epsilon^c) \tag{6}$$

This method enables us to focus on the most significant deviations between the estimated noise vectors, effectively filtering out less relevant information and ensuring that the editing process targets only the desired attributes while maintaining the integrity of the original video features.

## 5 EXPERIMENT

### 5.1 EXPERIMENT SETUP

**Dataset.** We evaluate the performance of our proposed GuideEdit on real-world videos sampled from the HDTF dataset (Zhang et al., 2021) and the VoxCeleb dataset (Nagrani et al., 2017). Specifically, we randomly select 20 videos from each dataset, ensuring diversity across gender, age, and skin tones. Each video consists of hundreds of frames, from which we randomly sample 32 consecutive frames for each evaluation. The selected frames are aligned and cropped following the approach in (Tzaban et al., 2022; Kim et al., 2023), and subsequently resized to a resolution of $256 \times 256$.

**Baseline.** We compare our method extensively with several previous state-of-the-art baselines. We choose diffusion-based editing method DVA (Kim et al., 2023) and transformer-based method Latent-trans (Yao et al., 2021). For GAN-based methods, we choose STIT (Tzaban et al., 2022), PTI (Roich et al., 2022) and StyleCLIP (Patashnik et al., 2021). Some of the baseline methods are designed for image editing, we adapt them into the video editing paradigm (the details can refer to Appendix C.1). It is important to note that, for a fair comparison of the reconstruction abilities of different editing methods, the original videos are used solely as input. None of the editing methods have access to the original videos during the output stage, ensuring that the reconstruction quality is evaluated independently of the input data.

**Metric.** For comprehensive evaluation of our proposed GuideEdit and the baseline methods, we utilize a range of evaluation metrics. For the evaluation of reconstruction performance, we use

SSIM (Wang et al., 2004), LPIPS (Zhang et al., 2018b), MSE and FID. For time consistency evaluation of manipulated videos, we apply TL-ID and TG-ID (Tzaban et al., 2022). For evaluating video editing performance, we use the Identity Preservation Rate (IPR), Target Attribute Change Rate (TACR) (Yao et al., 2021), and CLIP score. The attribute preservation rate measures the proportion of samples where non-target attributes remain unchanged during editing. The identity preservation score represents the average cosine similarity between the embeddings of the original frames and the manipulated results, reflecting how well the subject's identity is maintained. The CLIP score is computed based on the alignment between the target attribute and the edited frames.

**Implementation.** We implement the proposed GuideEdit using a diffusion autoencoder with a UNet architecture as the noise estimator. To enhance the model's ability to reconstruct the background in face videos, we fine-tune the pre-trained diffusion autoencoder from (Kim et al., 2023) on the HDTF dataset (the details of finetuning the diffusion autoencoder can refer to Appendix C.3). Note that during the editing process, the pre-trained diffusion autoencoder model remains frozen. We use the DDIM sampler, setting the inference time steps to 1000. The batch size for inference is set to 4, and all inference is performed on 4 RTX4090 GPUs.

## 5.2 RECONSTRUCTION EVALUATION

For video editing tasks, it is essential that the model can accurately reconstruct the original video from its encoded representation. To achieve this, we fine-tune the pre-trained diffusion autoencoder to enhance its ability to accurately reconstruct both the background and human face. We evaluate the reconstruction performance of GuideEdit against all baseline methods on the HDTF and VoxCeleb datasets, with the results reported in Table 1.

Table 1: The reconstruction performance of our GuideEdit and baselines on HDTF and Voxceleb datasets. The reported values are the mean of the averaged per-frame measurements for each video.

| Method | HDTF | | | | VoxCeleb | | | |
|---|---|---|---|---|---|---|---|---|
| | SSIM (↑) | LPIPS (↓) | MSE (↓) | FID (↓) | SSIM (↑) | LPIPS (↓) | MSE (↓) | FID (↓) |
| StyleCLIP | 0.6653 | 0.1984 | 0.0125 | 136.52 | 0.4830 | 0.3028 | 0.0183 | 233.60 |
| STIT | 0.5202 | 0.3978 | 0.0617 | 244.60 | 0.6669 | 0.2769 | 0.0513 | 179.27 |
| PTI | 0.6347 | 0.2476 | 0.0256 | 168.12 | 0.4737 | 0.3434 | 0.0337 | 227.43 |
| Latent-trans | 0.7035 | 0.1571 | 0.0068 | 137.70 | 0.6017 | 0.2208 | 0.0076 | 217.96 |
| DVA | 0.9448 | 0.0584 | 0.0003 | 33.531 | 0.9696 | 0.0130 | 0.0006 | 44.458 |
| GuideEdit | **0.9715** | **0.0108** | **0.0001** | **23.432** | **0.9779** | **0.0095** | **0.0004** | **24.840** |

Table 1 clearly demonstrates that our method achieves significantly better reconstruction performance compared to baseline methods on both the HDTF and VoxCeleb datasets. This highlights the superior ability of our model to faithfully reconstruct fine details in both the background and human face, underscoring its robustness and generalizability. We further provide a visual comparison of the reconstruction performance across different methods in Figure 6.

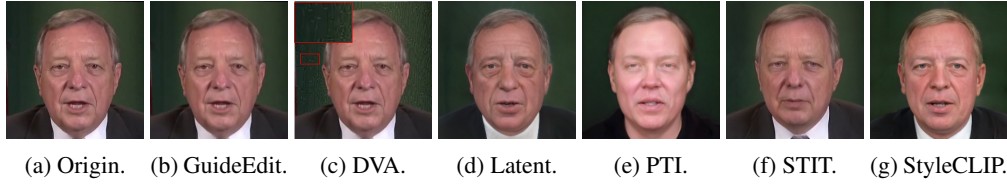

    (a) Origin.    (b) GuideEdit.    (c) DVA.    (d) Latent.    (e) PTI.    (f) STIT.    (g) StyleCLIP.

Figure 6: The comparison of the images reconstructed by our GuideEdit and five baseline methods with the original input image.

It can be seen from Figure 6 that baseline methods struggle to either preserve the identity of the characters or retain the background features. In contrast, our GuideEdit shows clear superiority in reconstructing the face videos, delivering more accurate restoration of both facial identity and background details. This enhanced reconstruction ability makes GuideEdit particularly effective for tasks where maintaining consistency between the original content and the edited results is crucial, highlighting its robustness in video manipulation.

## 5.3 EDITABILITY EVALUATION

### 5.3.1 QUANTITATIVE RESULTS

To thoroughly evaluate the editing capabilities of our proposed GuideEdit compared to baseline methods, we choose two general editing directions ("smile", "Mustache"). We compute and report the average values of key evaluation metrics, such as Identity Preservation Rate (IPR), Target Attribute Change Rate (TACR), and CLIP score, for both our method and the baseline approaches. The results, summarized in Table 2, illustrate how effectively each method handles these editing tasks, offering insights into their relative performance across different editing scenarios.

Table 2: The editing ability of our GuideEdit and baselines on HDTF and VoxCeleb datasets. The reported values are the mean of two editing directions ("Smile", "Mustache").

| Method | HDTF | | | | | VoxCeleb | | | | |
|---|---|---|---|---|---|---|---|---|---|---|
| | IPR (↑) | TACR (↓) | CLIP-Score (↑) | TL-ID (↑) | TG-ID (↑) | IPR (↑) | TACR (↓) | CLIP-Score (↑) | TL-ID (↑) | TG-ID (↑) |
| StyleCLIP | 0.8013 | 0.0329 | 0.7676 | 0.9997 | 0.9995 | 0.7051 | 0.0337 | **0.7670** | 0.9998 | 0.9993 |
| STIT | 0.8214 | 0.0341 | 0.7501 | 0.9866 | 0.9490 | 0.8131 | 0.0339 | 0.7383 | 0.9997 | 0.9994 |
| PTI | 0.7540 | 0.0327 | 0.7646 | 0.8238 | 0.8122 | 0.7140 | 0.0336 | 0.7627 | 0.7986 | 0.8047 |
| Latent-trans | 0.7515 | 0.0348 | 0.7450 | 0.9978 | 1.0000 | 0.7070 | 0.0335 | 0.7393 | 0.9999 | **1.0000** |
| DVA | 0.9244 | **0.0318** | 0.7685 | 1.0000 | 0.9977 | 0.8910 | 0.0341 | 0.7661 | 0.9999 | 0.9969 |
| GuideEdit | **0.9667** | 0.0338 | **0.7777** | **1.0001** | **1.0000** | **0.9033** | 0.0335 | 0.7607 | **1.0000** | **0.9999** |

As shown in Table 2, our proposed GuideEdit achieves the highest Identity Preservation Rate (IPR), highlighting its effectiveness in maintaining identity information during editing process. Additionally, our method demonstrates comparable temporal consistency to the baseline methods, further validating its robustness in preserving video quality over time.

### 5.3.2 QUALITATIVE RESULTS

We further provide the visualization of the manipulation videos of different editing methods in Figure 7. Due to the limit of space, only our method and three baseline methods are presented (the full comparison can refer to Appendix C.4).

As demonstrated in Figure 7, our method effectively edits the target attribute without impacting other facial attributes, ensuring that the character's identity remains intact throughout the editing process. Additionally, the background remains unaffected, showcasing the model's ability to localize changes specifically to the desired areas. This level of precision allows for high-quality edits while preserving both the identity of the subject and the original context of the scene, which is a significant challenge in FVE tasks.

To highlight the generalizability of our proposed method, we present the manipulation results of a single video across multiple editing directions in Figure 8. Our approach excels at handling highly intricate background details and dynamic scenes that include substantial head movements and speech—scenarios that typically challenge existing state-of-the-art methods. Furthermore, our method adeptly retains the stylistic elements of the original video, ensuring that the edited output blends seamlessly with the untouched portions. This results in an exceptionally natural appearance, with virtually no visible traces of editing. The ability to maintain such coherence across

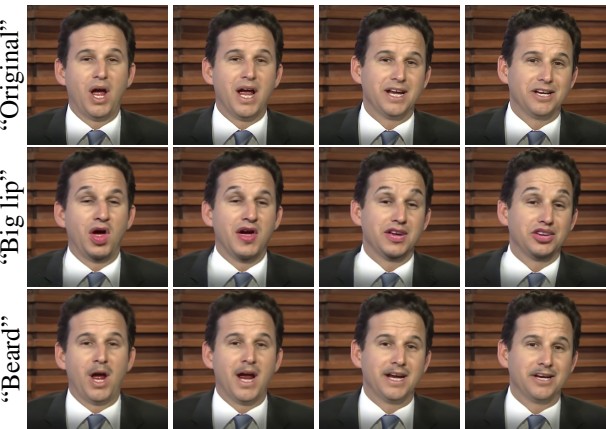

Figure 8: Manipulation results of our GuideEdit on a single video with two different editing directions: "Beard" and "Big Lip".

different editing tasks underscores the robustness and adaptability of our approach.

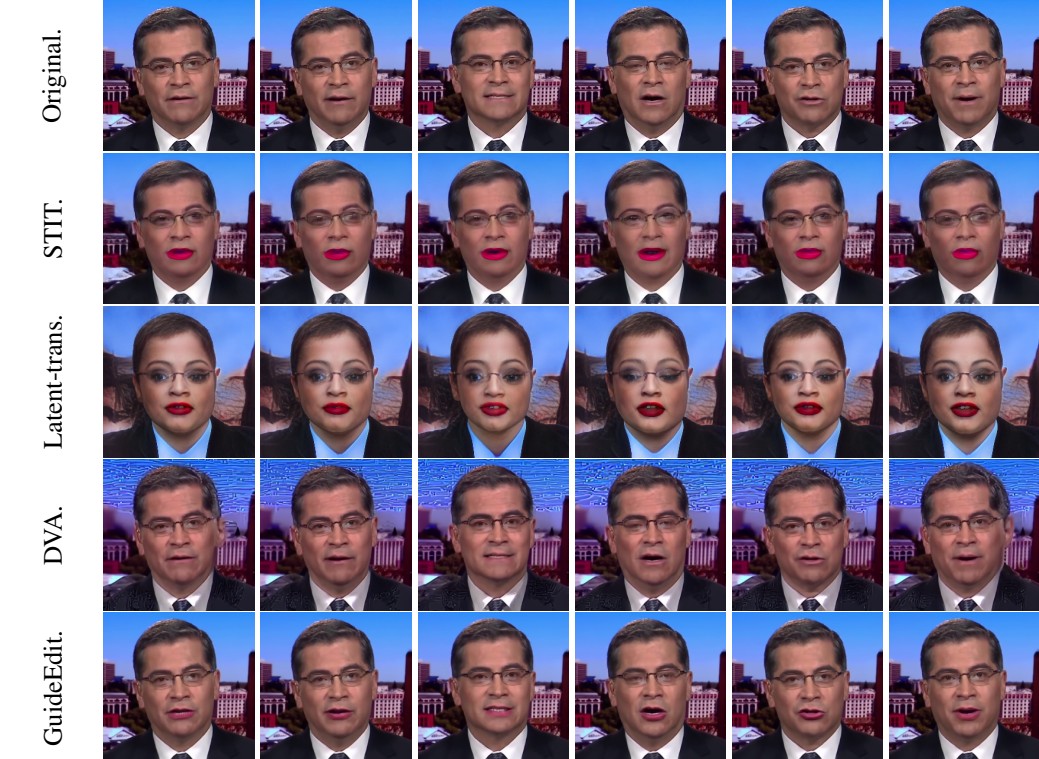

Figure 7: Comparison of editing performance of our GuideEdit to the previous video editing methods for editing direction 'Libstick'.

## 5.4 LATENT BASIS ANALYSIS

We extract the latent basis within the latent space as key indicators of the attributes. By calculating the similarity between the latent basis of the original video and the manipulated video under a specific editing direction, we can quantify the degree of change introduced during editing. This similarity metric serves as a guide for the editing process, enabling more precise adjustments and ultimately improving the overall quality of the edits.

In Figure 9, we present the change in similarity values at different denoising time steps for two editing directions: "Beard" and "Big Lip." The denoising time step ranges from 0 to T. As observed, the similarity is higher at larger time steps and lower at smaller time steps. This can be explained by the fact that at larger time steps, the latent space contains more noise, making the extracted latent basis of both the original and edited videos more similar.

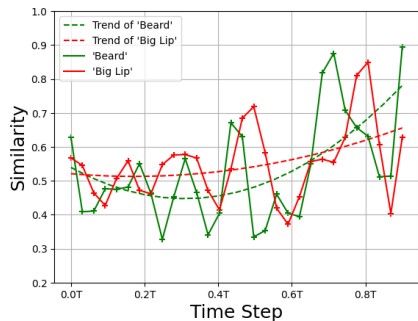

Figure 9: The similarity between the latent basis of the original video and the manipulated video evolves as the denoising process progresses.

In contrast, at smaller time steps, as less noise is present, the latent basis more accurately reflects the encoding features, leading to a greater distinction between the original and edited videos.

Furthermore, this observation aligns with the understanding that the model initially focuses on low-frequency signals during the early stages of the generative process, where the similarities between the original and edited videos are more pronounced. Over time, the model progressively shifts its attention to high-frequency signals, which highlight the introduced target attribute and the differences between the two videos. This result reinforces the common view of the coarse-to-fine behavior exhibited by diffusion models throughout the generative process (Kim et al., 2023).

## 5.5 ABLATION STUDY

We demonstrate the effectiveness of our proposed GuideEdit, and to gain deeper insights into the contribution of each component, we conduct an ablation study on different parts of GuideEdit. To assess the role of the latent basis, we remove its extraction and instead use the direct similarity of the latent space as a replacement. To evaluate the importance of proximal guidance, we perform experiments without applying it, isolating its impact on the overall performance.

The results of the ablation study for each component of GuideEdit are presented in Table3 and Figure 10. When the latent basis extraction is removed and the similarity of the latent variables is used as a replacement, the differences between the original and manipulated videos are not effectively highlighted. As a result, the video cannot be edited accurately, leading to a high Identity Preservation Rate (IPR) but a low Target Attribute Change Rate (TACR).

Table 3: The editing ability of our GuideEdit and baselines on HDTF and Voxceleb datasets. The reported values are the mean of two editing directions ("Smile", "Mustache").

| Method | IPR ($\uparrow$) | TACR ($\downarrow$) | CLIP-Score ($\uparrow$) | TL-ID ($\uparrow$) | TG-ID ($\uparrow$) |
|---|---|---|---|---|---|
| GuideEdit$_{w/o\ LBE}$ | 0.9831 | 0.0331 | 0.7437 | 0.9925 | 0.9775 |
| GuideEdit$_{w/o\ PG}$ | 0.8790 | 0.0337 | 0.7773 | 0.9770 | 0.8854 |
| GuideEdit | 0.9510 | 0.0329 | 0.7563 | 0.9986 | 0.9929 |

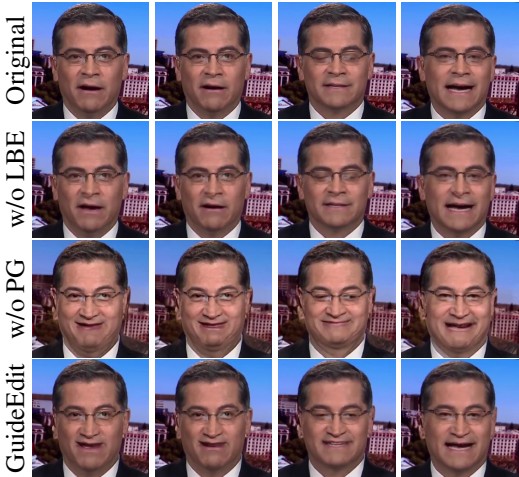

Figure 10: The ablation results of GuideEdit when apply editing direction:"smile".

Similarly, when the proximal guidance is removed, the model lacks direction for where the edits should occur, making it unable to identify the correct areas to manipulate. This results in the manipulated videos failing to preserve identity information, which is reflected in a low IPR and high TACR. The findings in Table 3 underscore the necessity of both the latent basis extraction and proximal guidance in improving the overall quality of video editing in our proposed GuideEdit.

We further present the visualization of the ablation study for GuideEdit in Figure10, where similar conclusions to those in Table 3 can be drawn. When the latent basis extraction (LBE) is removed, the video cannot be edited according to the specified attribute. This is because the key features associated with the encoded input are not properly highlighted, causing the editing degree to approach zero and resulting in a failure to apply the desired edits. On the other hand, when the proximal guidance (PG) is removed, the manipulated video fails to preserve identity features due to the absence of editing direction. These results emphasize the importance of each component of our proposed method in achieving successful and precise video editing.

## 6 CONCLUSION

In conclusion, we present GuideEdit, a novel diffusion-based method for FVE that effectively addresses the critical challenges of maintaining temporal consistency and preserving identity while manipulating specific attributes. Our approach leverages the inherent linearity of latent variables in the bottleneck layer of the diffusion U-Net model, enabling us to extract a latent basis that encodes key features related to target facial attributes. By comparing the latent basis of the original video with that of the manipulated video, we quantify the manipulation degree, which indicates the extent of changes made. This manipulation degree serves as a guidance for determining the specific components to be edited, allowing for fine-grained control at each denoising step. Integrating this precise control into the editing process enhances temporal consistency and ensures the preservation of identity, all while minimizing the introduction of artifacts. Extensive experiments conducted on diverse real-world videos demonstrate the effectiveness of GuideEdit, showcasing its ability to achieve precise, high-quality edits that maintain coherence across frames and preserve essential visual elements. This work not only advances the state of the art in FVE but also highlights the potential of diffusion-based methods for future generative modeling applications.

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

# A  ANALYSIS

## A.1  FORWARD DIFFUSION PROCESS

To help understand how the forward diffusion process changes the distribution of the video frames, We provide the changes of the frames when apply the forward diffusion process in Figure 11, the diffusion step ranges from 0 to 1000.

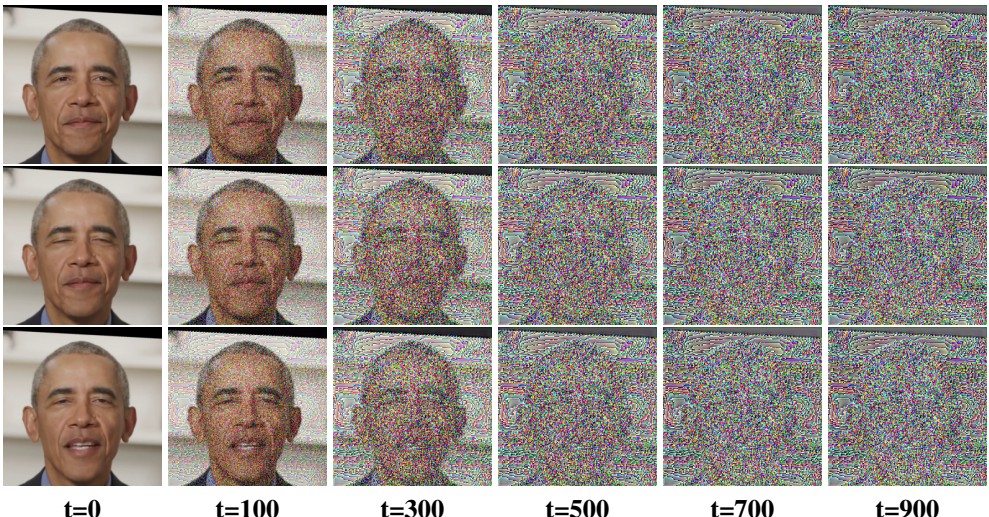

**t=0**  **t=100**  **t=300**  **t=500**  **t=700**  **t=900**

Figure 11: The illustration of the forward diffusion process, the diffusion step ranges from 0 to $T = 1000$.

## A.2  BACKWARD DENOISING PROCESS

To help understand the editing process of the diffusion-based model, we illustrate the editing process in Figure 12, the denoising time step ranges from 1000 to 0. It can be seen that the editing direction is integrated into the frames with the denoising process proceeds.

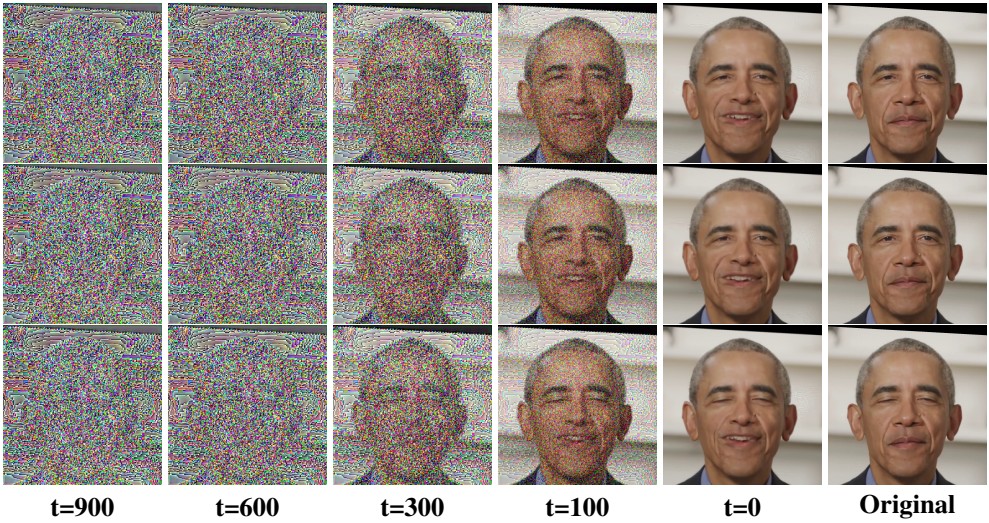

**t=900**  **t=600**  **t=300**  **t=100**  **t=0**  **Original**

Figure 12: The editing process that integrates the condition into the images during the denoising process, the denoising step ranges from $T = 1000$ to 0 and the editing direction is "smiling".

## B  VISUALIZATION

We demonstrate the editing performance of our proposed GuideEdit across multiple editing directions. Figure 13 presents the results for the editing direction "Beard," while Figure 14 highlights the performance for the editing direction "Smile." In Figure 15, we showcase the results for the editing direction "Eyeglasses," and Figure 16 illustrates the performance with the editing direction "Big Lip." These examples highlight the versatility and precision of GuideEdit in manipulating various facial attributes while maintaining the integrity of the original video.

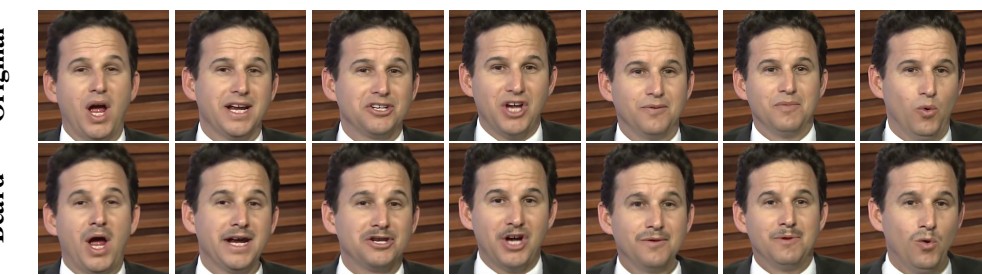

Figure 13: The editing of GuideEdit with editing direction "Beard".

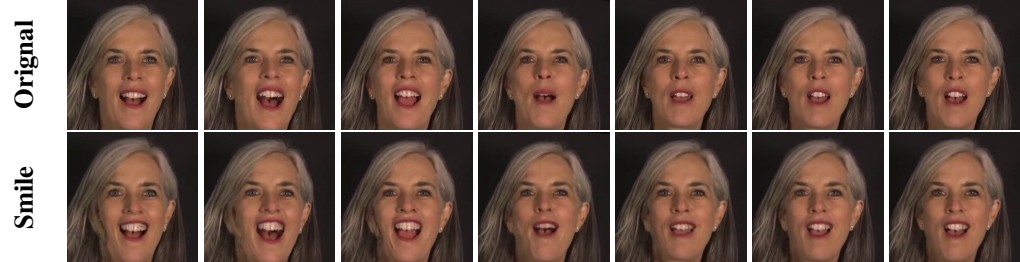

Figure 14: The editing of GuideEdit with editing direction "Smile".

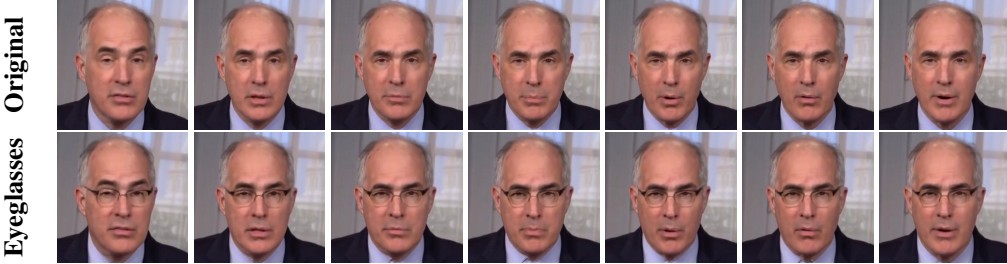

Figure 15: The editing of GuideEdit with editing direction "Eyeglasses"

## C  EXPERIMENT

### C.1  IMPLEMENTATION

We select several state-of-the-art methods for comparison: the diffusion-based editing method DVA Kim et al. (2023) and the transformer-based method Latent-trans Yao et al. (2021). For GAN-based methods, we include STIT Tzaban et al. (2022), PTI Roich et al. (2022), and Style-CLIP Patashnik et al. (2021).

It is important to emphasize that, for a fair evaluation of reconstruction capabilities, all methods only use the original videos as input. None of the methods have access to the original videos during

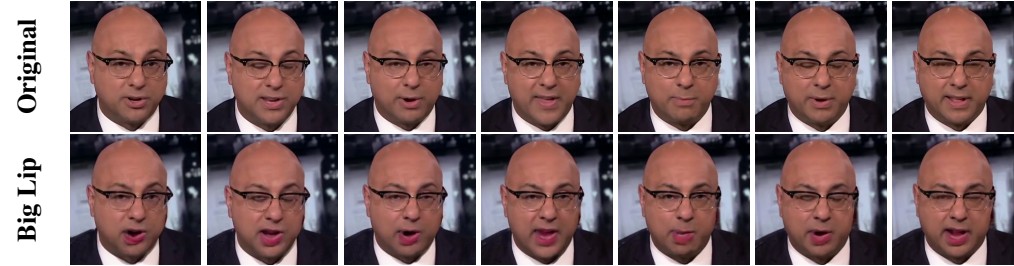

Figure 16: The editing of GuideEdit with editing direction "Big Lip"

the output generation phase, ensuring that the reconstruction quality reflects the true performance of each editing approach without reliance on the input data.

- DVA Kim et al. (2023): For the implementation of DVA, we use their CLIP-based editing method, and the editing scale $\alpha$ is set to 0.25 as recommended in their paper, and the input texts of the CLIP-based editing method are "Face" and "Face with *" for original video and the target manipulated video, other experiment settings are used the default settings.

- Latent-trans Yao et al. (2021): For the implementation of Latent-trans, we set the scaling factor $\alpha$ as 1.5 and the other settings are kept as recommended. And we use the output frames directly, the output frames are not blended with the original input frames.

- STIT Tzaban et al. (2022): We run edits with stitching tuning, and the edit ranges is set to 10101, the parameter $\beta$ is set to 0.2 and the *outer_mask_dilation* is set to 50. Other settings are kept as recommended. The output frames are used directly as well.

- PTI Roich et al. (2022): We use the default settings as recommended, and the frames of the videos are resized to 1024. We also use the output frames directly, without blending them into the original video frames.

- StyleCLIP Patashnik et al. (2021): We train the mappers of input videos with the default settinfs and use the attributes as the descriptions. Then we use the default settings to edit the videos and the output frames are used directly.

## C.2 OBTAIN CONDITION

To edit videos using diffusion-based models, the editing directions must first be mapped into conditions. We achieve this by leveraging the pre-trained CLIP model Radford et al. (2021) to encode the editing directions. In Section 4.1, we generate the original condition, denoted as $\mathcal{C}^r$ (see Equation 2), and represent the input with this original condition as $X_0^r$. The forward diffusion process is then applied to $X_0^r$ over the diffusion steps $\hat{T}$.

Next, the target conditions are initialized as $\hat{\mathcal{C}}^c = \mathcal{C}^r$. These target conditions are iteratively updated until the final conditions are obtained. At each diffusion step $t \in \hat{T}$, we compute the input $\hat{X}_t^c$ using the equation $\hat{X}_t^c = \mathcal{E}_i(X_t^0, \hat{\mathcal{C}}^c)$, ensuring that the editing directions are accurately incorporated into the denoising process.

The source text for $X_0^r$ is "face," and the target text is "face with $\delta$," where $\delta$ represents the target attribute. We use $I$ to denote the source text and $I_\delta$ to denote the target text. To quantify the difference between the source and target conditions, we utilize the CLIP loss function $\mathcal{L}_{clip}$ from Radford et al. (2021) to compute the loss. The loss function is formulated as:

$$\mathcal{L}_1 = \sum_{t=0}^{\hat{T}} \mathcal{L}_{clip}(I, X_t^r, I_\delta, \hat{X}_t^c) \tag{7}$$

This loss helps guide the model toward generating video frames that align with the target attributes defined by $\delta$.

Then to keep the consistency of the background information of the reconstructed frames under the target conditions with the original video frames, another loss function is used:

$$\mathcal{L}_2 = \frac{1}{\hat{T}} \sum_{t=0}^{\hat{T}} (X_t^r, \hat{X}_t^c) \tag{8}$$

and to control the updated conditions don't vary too much:

$$\mathcal{L}_3 = 1 - \frac{\mathcal{C}^r \hat{\mathcal{C}}^c}{||\mathcal{C}^r|| ||\hat{\mathcal{C}}^c||} \tag{9}$$

then the optimization object can be obtained as:

$$\mathcal{L} = w_1 \mathcal{L}_1 + w_2 \mathcal{L}_2 + w_3 \mathcal{L}_3 \tag{10}$$

where $w_1, w_2, w_3$ are constants. And through minimizing $\mathcal{L}$ until convergence, we could get the trained conditions $\Delta_c = \mathcal{C}^r - \hat{\mathcal{C}}^c$.

**Settings for Obtaining Conditions**

In this paper, we use the pre-trained CLIP model, specifically the ViT-B/32 version. The weights $w_1, w_2, w_3$ are set to 5, 1, and 3, respectively, and the forward time step $\hat{T}$ is set to 5. The learning rate is set to 0.002, with a batch size of 1 during training. The number of updating steps is fixed at 1000.

## C.3 FINETUNE DIFFUSION AUTOENCODER

We finetune the pre-trained diffusion autoencoder from Kim et al. (2023) on the HDTF dataset. The loss function used for finetuning consists of two components. The first component is the standard DDIM (Denoising Diffusion Implicit Models) loss function, represented as:

$$\mathcal{L}_{ddim} = \mathbb{E}_{\epsilon_t \sim \mathcal{N}(0,I)} ||\epsilon_t^r - \epsilon_t||_1 \tag{11}$$

where $\epsilon_t^r$ can refer to Equation 3 and $\epsilon_t$ is the true noise, $t \in T$. This loss is minimized to ensure accurate denoising and reconstruction during the finetuning process.

To enhance the robustness of the model to noise, we sample images given the time step with two different noise realizations, denoted as $\epsilon_1$ and $\epsilon_2$, where $\epsilon_1, \epsilon_2 \sim \mathcal{N}(0,1)$. The sampled images are represented as $\hat{X}_t^1$ and $\hat{X}_t^2$.

The loss function for this sampling process can be formulated as follows:

$$\mathcal{L}_{reg} = \mathbf{E}_{\epsilon_1, \epsilon_2 \sim \mathcal{N}(0,1)} ||\hat{X}_t^1 - \hat{X}_t^2||_1 \tag{12}$$

This loss encourages the model to accurately predict the noise for both sampled images, thereby improving its robustness against variations in noise during the denoising process.

The final optimization objective for finetuning the diffusion autoencoder is $\mathcal{L} = \mathcal{L}_{ddim} - \mathcal{L}_{ddim}$

**Settings for Finetuning the Diffusion Autoencoder**

We finetune the diffusion model on HDTF dataset. The learning rate is set to 1e-4 and the dropout is set to 0.1, and we sample from each videos 16 frames during each training step. The batchsize is set to 16, the total training steps is set to 120000. And we set the seed to 0, the diffusion step $T = 1000$. The experiment is performed on 4 RTX4090 GPUs.

## C.4 ADDITIONAL RESULTS

The full comparison of our proposed GuideEdit and the baseline methods is presented in Figure 7 with editing direction "Lipstick", and Figure 18 with editing direction "Smile".

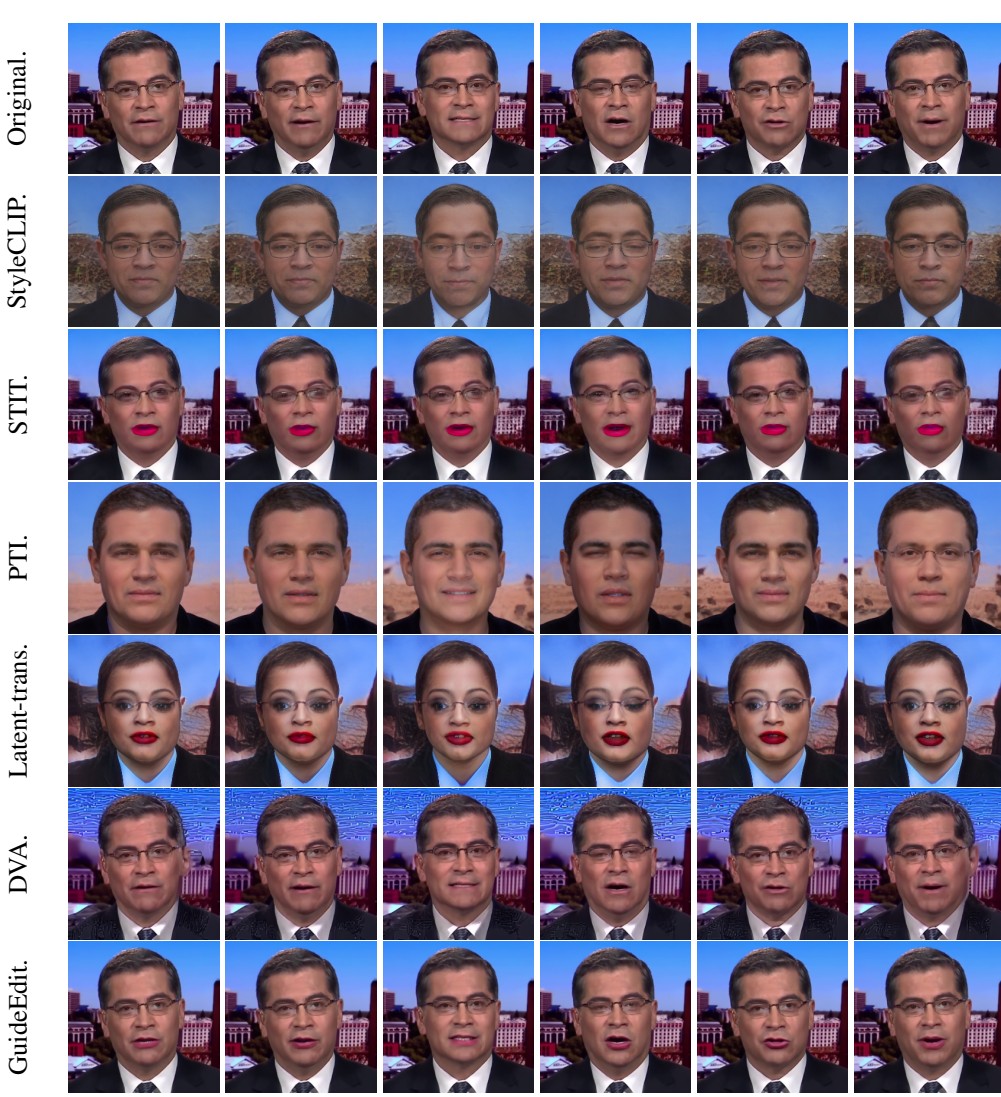

Figure 17: Comparison of editing performance of our GuideEdit to the previous video editing methods for editing direction 'Libstick'.

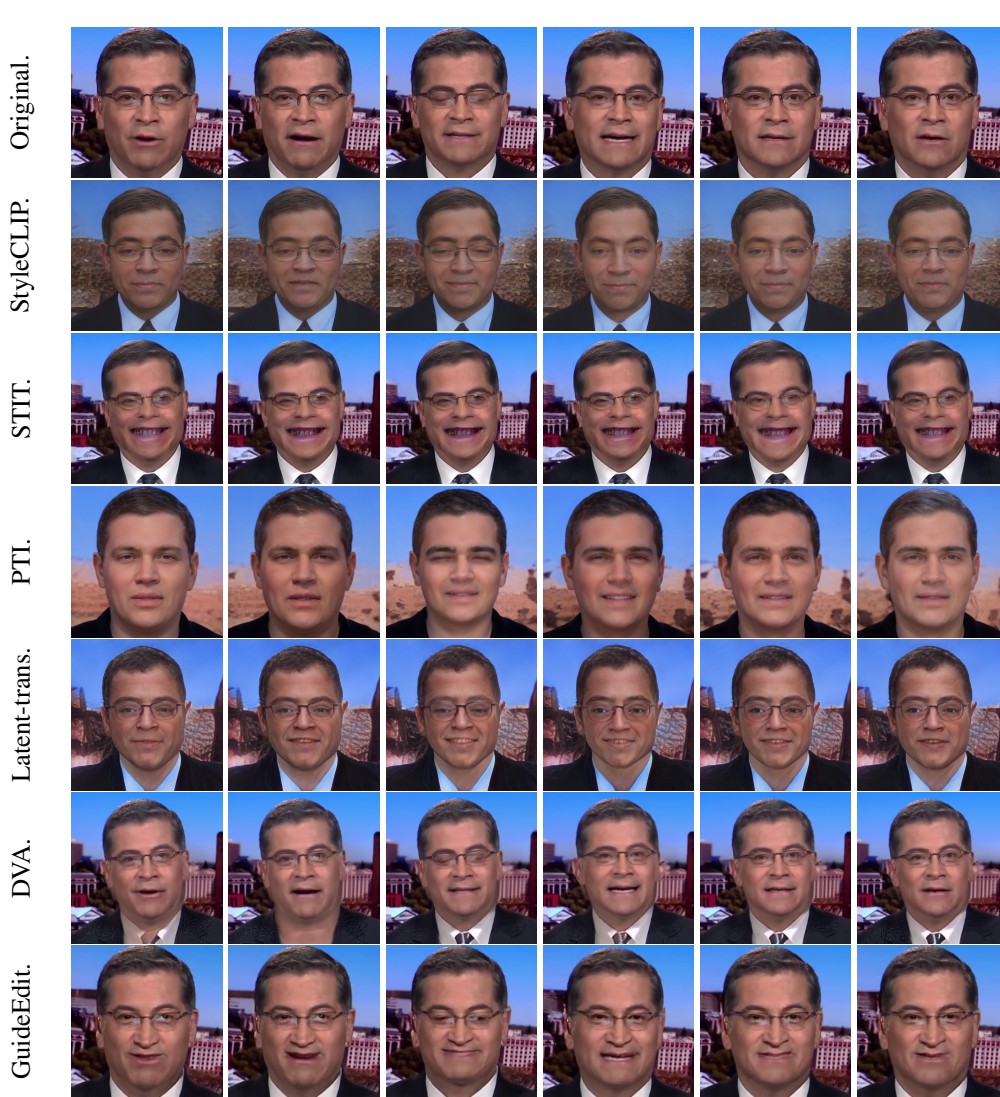

Figure 18: Comparison of editing performance of our GuideEdit to the previous video editing methods for editing direction 'Smile'.

