# OpenReview forum: "GuideEdit: Enhancing Face Video Editing with Fine-grained Control"
_ICLR.cc/2025/Conference — ICLR 2025 Conference Withdrawn Submission_

### Official Review · Reviewer_bDjK · 2024-10-27

**Soundness:** 2
**Presentation:** 3
**Contribution:** 2
**Rating:** 5
**Confidence:** 4

**Summary:**

This paper proposes GuideEdit that guides fine-grained controlled editing by manipulation of targeted attributes in a latent space assuming under the linear property between input frame and targeted attributes.

**Strengths:**

[1] Proposed approach is based on conceptual feature engineering, which affects many applicability of diffusion-based multi modality.

[2] Figure is illustrative and solid experiments are performed.

**Weaknesses:**

[1] This study relies on the local linearity of the latent variable, but the theoretical foundation depends almost entirely on citations from other studies. Since this concept is central to the research, it needs to be thoroughly explained to ensure reader comprehension. Without this, the proposed method lacks persuasive power.

[2] The method description is overly abstract, and the mathematical notation lacks foundational clarity, making it difficult to follow.

[2-1] Specifically, in Section 4.1, what are $X$ and $X^{c}_{0}$.

[2-2] Numerous subscripts and superscripts are used, but their meanings are not adequately explained, which detracts from the understanding of the equations.

[2-3] The mathematical symbols are also not used appropriately. The ':' notation is typically for mappings, but here it is mixed with function definitions, leading to confusion.

[2-4] Additionally, the mappings are incorrectly defined, as mappings should connect one space to another. Notably, $\mathcal{X}$, $\mathcal{C}$, $\mathcal{H}$ are not spaces.

[2-5]  Furthermore, $T_{C}$ is defined as a space but is inconsistently mixed with set notation.

**Questions:**

My questions are based on weakness above.

---

### Official Review · Reviewer_oHKD · 2024-11-02

**Soundness:** 2
**Presentation:** 3
**Contribution:** 2
**Rating:** 3
**Confidence:** 4

**Summary:**

This paper proposes a SD inversion-based face video editing method that manipulates the latent basis in the SD latent space to edit the input face video. Specifically, the authors assume the inherent linearity of the latent variables in the pretrained U-Net, and then utilize the difference between the input video and the manipulated video as guidance for specific editing. The authors believe that high-quality results are presented, but the resulting videos contain obvious unacceptable artifacts, are unstable, and the generated portrait identity may be affected.

**Strengths:**

1. The paper is well-written; exploring the decoupling of latent features in SD for editing is interesting to me.
2. The proposed method is easy to follow, and the denoiser network does not require training for editing.
3. The paper introduces some analytical experiments, including the similarity between the latent basis of the original video and the manipulated video at different time steps.

**Weaknesses:**

1. Although the motivation and theoretical analysis in the method design are somewhat reasonable, the edited face videos exhibit noticeable artifacts, and the facial expressions and IDs are affected by the modified lip color. This raises doubts about the authors' assumption that the inherent linearity of the latent variables in the pretrained U-Net is suitable for face video editing.
2. The videos presented in the paper demonstrate significant inter-frame instability. Since the authors use image-based SD inversion, they have not implemented any additional algorithms to address face video editing.
3. While proximal guidance utilizes the feature basis similarity between the manipulated video and the original video, the actual implementation only uses it to calculate a binary mask. This implementation is inconsistent with the preceding theoretical analysis.
4. In the implementation section, the DDIM sampler uses 1,000 steps, and both the manipulated video and the original video require inversion, which severely slows down the inference speed of the method.
5. Most values for TG-ID and TL-ID in Table 2 are greater than 0.99, which raises questions about the validity of using these two metrics to assess temporal stability.
6. In the ablation experiments of the paper, multiple indicators show that the method performs better in terms of image quality after editing when the LBE or PG module is not included.

**Questions:**

1. The cases presented in the paper involve local editing. How would the method perform if applied to larger-scale edits, such as hairstyle or gender changes?
2. Why does the paper use a 1,000-step DDIM, and what is the difference between this and DDPM?
3. What are the results of the reconstructed videos after manipulating using GuideEdit?

---

### Official Review · Reviewer_fd4W · 2024-11-02

**Soundness:** 2
**Presentation:** 3
**Contribution:** 2
**Rating:** 5
**Confidence:** 4

**Summary:**

In this paper, the authors propose GuideEdit, a method designed to address the challenge of face editing while preserving identity consistency. The approach leverages the idea that U-Net models in diffusion frameworks have a bottleneck layer, which can be approximated as a linear transformation between the input and feature representations. Building on this, the authors introduce a technique to manipulate the diffusion process specifically through this bottleneck layer to achieve controlled editing.

**Strengths:**

- The method’s use of inherent linearity in diffusion models for precise editing on video frames is a notable strength. I appreciate how it leverages this natural characteristic of diffusion models, effectively harnessing their underlying architecture.

- The method outperforms previous baselines in quantitative evaluations and achieves superior visual quality in the provided figures. Additionally, the ablation study effectively demonstrates the contribution of each component to the overall performance.

**Weaknesses:**

- I am curious why a model based on video editing with inpainting could not effectively address this task. For instance, to add eyeglasses, a two-stage baseline could operate as follows: (1) generate masks for each frame using an off-the-shelf method (e.g., SAM), and (2) apply methods such as AVID: Any-Length Video Inpainting with Diffusion Model (CVPR 2024) or Videoshop: Localized Semantic Video Editing with Noise-Extrapolated Diffusion Inversion (ECCV 2024). Could you elaborate on the potential differences between such a baseline approach and your method?

- It appears that Section 4.2 draws from concepts described in the paper "Understanding the Latent Space of Diffusion Models through the Lens of Riemannian Geometry" (NeurIPS 2023), as cited. While this section is presented as a component of GuideEdit, I could not identify significant differences in the approach to latent basis extraction between your description and that of the referenced paper. Could you clarify any novel aspects in your latent basis extraction method compared to the work outlined in the NeurIPS 2023 paper?

- The task is defined as "face video editing with precise control," but it would be helpful to understand more specifically which attributes can be controlled using this approach. Could you clarify the types of features or attributes that can be adjusted and to what extent this method allows for precise control? This information would provide a clearer picture of the method’s capabilities and limitations in achieving detailed adjustments.

**Questions:**

- have a question regarding frame-by-frame processing: from my understanding, the frames are processed independently. What mechanisms or strategies ensure that the desired edits remain consistent across frames (e.g., that generated eyeglasses retain the same style and appearance throughout the video)?

- I am also curious about the runtime of this approach for videos of a specific duration. Understanding the computational demands would provide insights into the method’s efficiency.

- I would appreciate more video results, as the supplementary material currently includes only a limited selection. Additional examples would help further demonstrate the approach’s effectiveness and consistency in various scenarios.

---

### Official Review · Reviewer_7sEg · 2024-11-02

**Soundness:** 2
**Presentation:** 3
**Contribution:** 2
**Rating:** 3
**Confidence:** 4

**Summary:**

The paper proposes a method namely GuideEdit which is a video editing framework using Diffusion models. In the the paper, the authors claim to achieve precise control by leveraging the local linearity of the latent variables in the bottleneck layer of the UNet architecture.
The proposed method calculates the  similarity between the latent basis of the original and edited video, and hence the degree of modification is computed. The basis are also used to apply the attribute changes to only intended attributes. The authors show examples of edited videos and comparisons with StyleGAN counterparts.

**Strengths:**

1) The paper is able to use the prior of the diffusion models and the bottleneck to perform edits to the original video. The basis extracted by the method can be used to focus on a specific attribute for video editing.

2) The paper shows multiple examples of video editing. The editing is performed on multiple identities and people with different ages.

3) The paper compares with the StyleGAN based methods which are known to have a smooth latent space enabling video editing.

**Weaknesses:**

1) I am not impressed with the quality of the videos produced in the general. The quality of videos shown in the paper is low. In the videos there are some unnatural artifacts, related to lips (eg figure 1) and eyeglasses (in the suppl.).

2) The examples provided in the suppl. are limited. It is not clear how well the method works. Without looking at many examples produced by the method, it is difficult the assess the quality and the scope of the method.

3) The paper only shows specific edits, with minor modifications.

4) There are no pose variations in the identities and the selected examples are limited.

**Questions:**

1) How to avoid the unnatural artifacts ?

2) I would like to see more examples in the supplementary. Is there a reason why only few examples are shown?

3) Why are only few edits shown in the paper ? What are the failure cases?

4) How does the method perform on difficult cases of extreme poses and lighting variations?

---

### Note · Authors · 2024-11-14

**Comment:**

Thanks for the efforts of all the reviewers, we will take these valuable comments into consideration to improve our paper!

**Withdrawal Confirmation:**

I have read and agree with the venue's withdrawal policy on behalf of myself and my co-authors.